# Earth system instability amplified biogeochemical oscillations following the end-Permian mass extinction

Zi-Heng Li [1,2], Timothy M. Lenton [2], Fei-Fei Zhang [3], Zhong-Qiang Chen [1] ✉ & Stuart J. Daines[4]

After the end-Permian mass extinction, the Earth system underwent extreme ecological and environmental fluctuations, including high temperatures, recurrent oceanic anoxia, and carbon cycle oscillations as demonstrated by the geochemical isotope proxy records. However, the underlying mechanism behind these oscillations remains poorly understood. Here we propose that they were produced by a coupled oscillation mode of marine phosphorus (**P**) and atmosphere–ocean carbon (**A**), driven by nonlinear redox controls on marine phosphorus burial. Our modeling demonstrates that the initial emplacement of the Siberian Traps and the mass extinction (on land and in the ocean) directly led to an early Triassic greenhouse. More importantly, it homogenized the ocean floor redox condition towards anoxia, activating amplifying feedbacks and destabilizing the system. The internal dynamics of an unstable system—rather than recurrent volcanic shocks—triggered the periodic oscillations (limit cycles) of serial excursions in carbonate carbon and uranium isotopes during the early Triassic.

The end-Permian mass extinction (EPME, 251.9 Myr ago) caused a pronounced biodiversity crisis and a protracted recovery of both marine and terrestrial ecosystems[1-4]. It also witnessed dramatic and periodic biogeochemical oscillations, which recurred during the early Triassic (Fig. 1). These include drastic carbon cycle oscillations indicated by large carbonate carbon isotopic ($\delta^{13}C_{carb}$) excursions[5], multiple recurrences of widespread oceanic anoxia evidenced by carbonate uranium isotopes ($\delta^{238}U_{carb}$[6,7]) and sustained hothouse regimes elucidated by phosphatic oxygen isotopes ($\delta^{18}O_{apatite}$[8]). The spatial-temporal patterns of these environmental perturbations, accompanied by biotic rebounds (i.e., Guiyang biota[9]) and the Smithian-Spathianboundary extinction[10] during the early Triassic, have been widely documented. Nevertheless, the underlying mechanism driving this unusual biogeochemical cycling during the early Triassic hothouse is unclear.

Mass balance models have been used to decipher the initial negative $\delta^{13}C_{carb}$ excursion across the Permian–Triassic (P–Tr) boundary. Triggered by the eruption of the Siberian Traps[11], the combination of large injections of multiple sources of isotopically light carbon into the ocean-atmosphere system, and an overall reduction in global organic matter burial is thought to be responsible for the end-Permian $\delta^{13}C_{carb}$ excursion and high atmospheric $CO_2$ level[12,13]. The collapse of terrestrial ecosystems and oxidation of terrestrial biomass could have facilitated the earliest Triassic $\delta^{13}C_{carb}$ excursion (252–251.8 Ma[14]). For the longer timescale, several pulses of massive volcanic activities are suggested to be responsible for the early Triassic carbon cycle oscillations[15].

[1]State Key Laboratory of Geomicrobiology and Environmental Changes, China University of Geosciences, Wuhan, China. [2]Global Systems Institute, University of Exeter, Exeter, UK. [3]State Key Laboratory of Critical Earth Material Cycling and Mineral Deposits, School of Earth Sciences and Engineering, and Frontiers Science Center for Critical Earth Material Cycling, Nanjing University, Nanjing, China. [4]University of Exeter, Exeter, UK. ✉e-mail: zhong.qiang.chen@cug.edu.cn

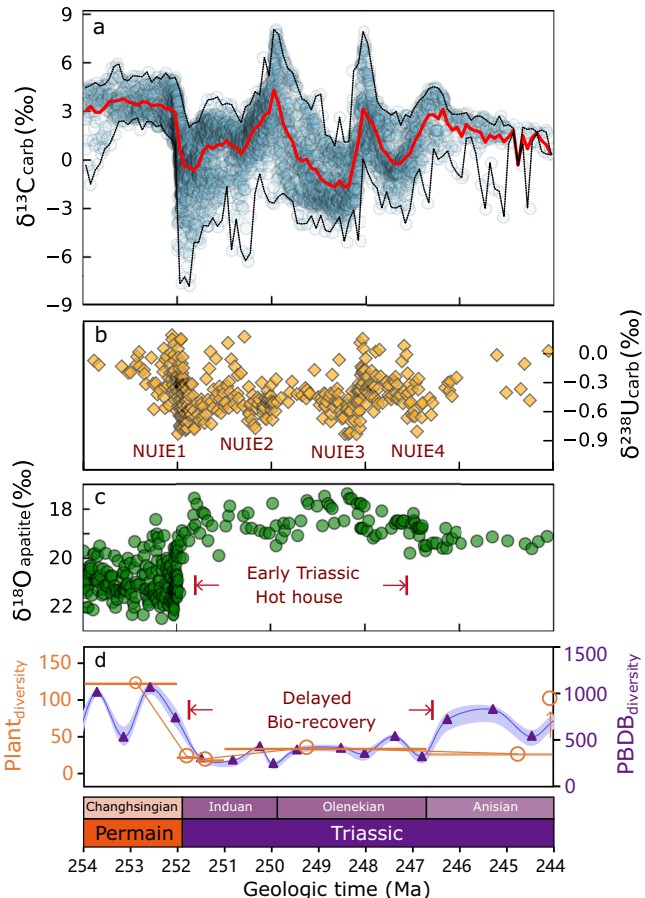

**Fig. 1 | Environmental perturbations and delayed recovery of global marine and terrestrial biodiversities after EPME.** See Supplementary Information (Table S1) for datum sources, or find data from Supplementary Dataset 1. **a** Marine carbonate carbon isotope ($\delta^{13}C_{carb}$) profile. Red and black lines are the moving average, minimum, and maximum with a moving window = 0.1 Myr. **b** Marine redox proxy, carbonate uranium isotope ($\delta^{238}U_{carb}$) profile. Four negative Uranium isotope excursions (NUIE) at around 252 Ma, 250.5 Ma, 248.5 Ma, and 247 Ma represent multiple expansion peaks of ocean anoxia. **c** Sea surface temperature proxy, apatite oxygen isotope ($\delta^{18}O_{apatite}$) profile, showing the early Triassic hothouse regimes (252–247 Ma). **d** End-Permian to late early Triassic terrestrial plant and marine invertebrate (from PBDB) diversities, showing delayed biotic recovery[4,70].

In this forced-response framework[16], the ocean-atmosphere carbon cycle linearly responds to the physical forcing (e.g., volcanism $CO_2$ input). Or conversely, the carbon input flux and rate can be inferred from the $\delta^{13}C_{carb}$ records through inverse modeling[13,17]. However, the age model constraints on the Siberian Traps only support a relatively short active period during ~252.2 Ma to 251 Ma[11,18], which leaves the early Triassic lacking a comparable magnitude of light carbon source[19] to trigger the repeated $\delta^{13}C_{carb}$ excursions observed (Fig. 1a). Moreover, mercury anomaly is considered a reliable proxy indicating large igneous province (LIP) volcanism during the P–Tr transition[20,21]. However, extensive studies show that mercury anomalies are only confined to the P–Tr transition, and are not recorded in the rest of the early Triassic successions, failing to support occurrences of LIP volcanism during that time[22]. This means another explanation for the ~8 Myr of extreme biogeochemical oscillations (Fig. 1) needs to be sought.

An alternative explanatory framework is that the biogeochemical cycles in Earth's land-atmosphere–ocean system (hereafter, "the system") contain several mechanisms via which amplifying nonlinear feedbacks could lead to instability[16,23,24]. It is already recognized that nonlinear redox controls on marine phosphorus burial[25] can generate limit-cycle oscillations in the phosphorus and oxygen cycles (**P–O**) with

a characteristic timescale of ~5–8 Myr. Similar models have been employed to interpret apparent periodicity in Cretaceous oceanic anoxic events (OAEs)[26]. Independently, during the Cretaceous Thermal Maximum at 97–91 Ma, feedbacks in the coupled nitrogen, iron, phosphorus (**N–Fe–P**) cycles have been used to explain periodic iron speciation oscillations with a timescale of ~40 kyr[27]. In this model, restricted lateral water exchange results in the oscillation between iron-rich and sulfidic (euxinic) states. Likewise, Alcott et al.[28] suggested that instability at intermediate oxygen levels during the progressive oxygenation of the Earth system through the Neoproterozoic–Paleozoic transition could explain extreme variability in atmosphere–ocean oxygen in that time. However, such a characteristic oscillation period (set by the atmosphere–ocean oxygen, **O** timescale) is too long to interpret the cyclicity of the early Triassic excursions within 10 Myr (Fig. 1). More recently[29], a dynamical system analysis of the coupled **P–O–A** biogeochemical cycles has demonstrated the possibility of faster **P–A** oscillations of Neoproterozoic–Paleozoic phosphorus and pCO2 levels, with a characteristic timescale of ~2–3 Myr. This implies that internal feedbacks and oscillations may be a plausible mechanism for the early Triassic excursions in $\delta^{13}C_{carb}$ and $\delta^{238}U_{carb}$.

In this alternative forced-stability-response view, the same external forcing can produce different responses if the underlying system is in a different stability state. Given that the end-Permian mass extinction and rapid injection of greenhouse gases have long been suggested to destabilize the system[30], we attempt to quantify how the environmental perturbations destabilized the system after the EPME, ultimately tipping the recurrent biogeochemical oscillations during the early Triassic.

We approach the problem using dynamical systems analysis of a model for the coupled biogeochemical cycles of phosphorus, carbon, and oxygen[29], which can capture different stable and unstable system configurations. We test the effect of previously suggested external forcings across the end-Permian to early Triassic interval on these configurations. The forcings represent the impact of the emplacement of the Siberian Traps[31] and the collapse of terrestrial ecosystems[31,32]. Model-data comparisons of multiple proxies are employed to evaluate the influence of variations in system stability on the secular changes of the carbon cycle, ocean redox condition, and seawater temperature in the early Triassic hothouse world.

## Results and discussion
### Key processes leading to system destabilization
The understanding of key controls on the long-timescale coupled biogeochemical cycles of marine phosphorus (**P**), atmosphere–ocean oxygen (**O**), and carbon (**A**)[25–27] is crucial in reconstructing the processes destabilizing the early Triassic Earth system (Fig. 2). Based on the typical geochemical view of oceanic nutrient-limitation[24,33], we assume the concentration of **P** is the ultimate limiting nutrient for primary production. Other nutrient limitations e.g., marine nitrogen (**N**) and iron (**Fe**), have much shorter residence times in the ocean compared to **P**, hence, on the ~1 Myr timescale, N and Fe content adjust quickly and can be considered as in source-sink balance all the time. This is illustrated by our extended model run with coupled **P–O–A** plus **N** cycles. We assume that the nitrogen fixation rate is greatly enhanced during ocean anoxia to balance the enhanced denitrification (Supplementary Information and Fig. S15). In which, **N** is the proximately limiting nutrient, but **P** controls the dynamics of the system on geological timescales. In addition, Fe limitation was more likely to have occurred in the Late Permian, rather than the early Triassic[34].

Following previous approaches[35,36], we assume that the marine phosphorus reservoir is controlled by the balance between weathering and burial. Atmosphere–ocean carbon is controlled by the balance among degassing, oxidative weathering and carbonate weathering input, and burials of organic carbon and carbonate. The oxygen

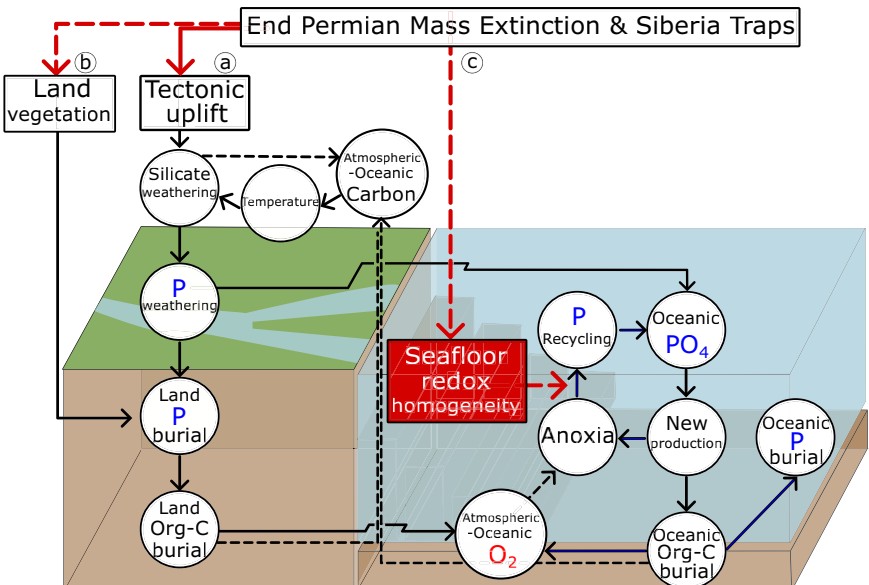

**Fig. 2 | Feedback diagram illustrating the plausible key processes leading to the system's destabilization.** Solid and dashed arrows show positive and negative effects, respectively. The boxes show three key processes that contribute to the system's destabilization. **a** The emplacement of the Siberian Traps resulted in elevated tectonic uplift, which subsequently enhanced silicate weathering and then phosphorus weathering. **b** The collapse of the terrestrial ecosystem halted the burial of phosphorus and organic carbon on land. **c** The collapse of marine ecosystems, combined with the above two processes, led to a homogenization of ocean floor redox conditions towards an anoxic state, and facilitated the anoxia-phosphorus-burial feedback. The latter includes detailed positive and negative feedback loops within redox-sensitive phosphorus cycles[26]. The positive feedback loop highlights how the elevated oceanic phosphate concentrations stimulate new production, leading to anoxia and the recycling of phosphorus from sediments, effectively trapping the system in anoxic conditions, with a timescale of ~0.1 Myr. The negative feedback mechanism, characterized by enhanced organic carbon burial, results in oxygen accumulation over a longer timescale of ~1 Myr, alleviating marine anoxia.

reservoir is controlled by the balance between the net oxygen source from organic carbon burial, and sinks from oxidative weathering and degassing of reductant input. The system dynamics are represented by three equations (Table S3), and can be visualized in three-dimensional phase space (Fig. 3). P weathering has three main contributors: silicate (80%), carbonate (14%), and oxidative (6%) weathering[36]. Here we made the simplification that the P weathering is 100% controlled by silicate weathering, as previous tests indicate that varying contributions from silicate, carbonate, and oxidative weathering have a minor impact on the overall ***P–O–A*** dynamics[29]. Given the relatively short timescale compared to the cycling of sedimentary rock reservoirs, the degassing fluxes are fixed at constant values (Table S4).

The emplacement of the Siberian Traps and the EPME plausibly destabilized the system through three key processes (Fig. 2).

First, the degassing and Siberian Trap volcanism increased atmosphere–ocean $pCO_2$ levels (***A***) and together with associated global warming intensified silicate, carbonate, and oxidative weathering to enhance the input of ***P*** to oceans, leading to an early Triassic system with elevated oceanic phosphate ($PO_4$) concentrations[31].

Second, the collapse of terrestrial ecosystem (de-vegetation[4]) reduced the burials of phosphorus and organic carbon derived from the land, lowering atmosphere–ocean oxygen ($O_2$) levels, and making the system reliant on a single long-term source of oxygen from the burial flux of marine organic carbon ($C_{org}$), rendering the coupled ***P–O–A*** cycles more sensitive to variations in that source. The globally integrated marine organic carbon burial flux depends on the marine phosphorus burial flux, its local redox sensitivity (expressed as the local $C_{org}:P_{total}$ ratio[37]), and the degree of homogeneity of ocean floor redox conditions[29]. There are two competing effects on the global integrated marine phosphorus burial flux as marine phosphorus level increases: increased marine productivity (new production, Fig. 2) leads to increased phosphorus burial, whereas increased water-column oxygen demand leads to increased ocean floor anoxia and phosphorus

recycling (Fig. 2). Which effect dominates depends on the homogeneity of ocean floor redox conditions.

Third, following established models[38,39], we assume that the oxygen levels of bottom waters are controlled by the balance between oxygen supply—through the transport of surface waters equilibrated with the atmosphere—and oxygen demand, which is influenced by the supply of limiting nutrients and the efficiency of nutrient uptake[36]. Nutrient uptake efficiency is controlled by a parameter ($k_u$). In the present-day ocean, the globally integrated oxygen supply exceeds the oxygen demand. After the EPME, this supply-demand balance was seriously disrupted. The phosphorus weathering flux and oceanic phosphorus concentration were likely increased, while the hot-house regime reduced the oxygen solubility of surface waters and potentially increased the remineralization of organic matter[40]. These processes heightened oxygen demand. Meanwhile, the removal of land $C_{org}$ burial and increased oxidative weathering diminished oxygen supply.

Moreover, previous studies highlighted the crisis of the eukaryotic marine phytoplankton and the bloom of cyanobacteria during the early Triassic: (1) unambiguous decline in eukaryotic marine phytoplankton is evidenced by disappearance of the e.g., Dasyclad algal during the whole early Triassic[1,41]; (2) a general increase in the abundance of green sulfur and N-fixing cyanobacteria is evidenced by high 2-methylhopanoid indices in basal Triassic shales[42–45]; and (3) the bloom of the cyanobacteria and other microbes is also evidenced by the widespread distributions of microbialites (stromatolites, thrombolites and other forms), and broad "anachronistic facies" or "microbially induced sedimentary structures (MISS)" (including oncoids, giant ooids, microbial mats, sand veins, wrinkle structures, vermicular limestone, flat-pebble conglomerates, cement fans etc.) worldwide[46–53].

Crucially, smaller-celled phytoplankton are more efficient at nutrient uptake thanks to a greater surface-area-to-volume ratio, as uptake is primarily diffusion-limited under light-saturated conditions[54,55].

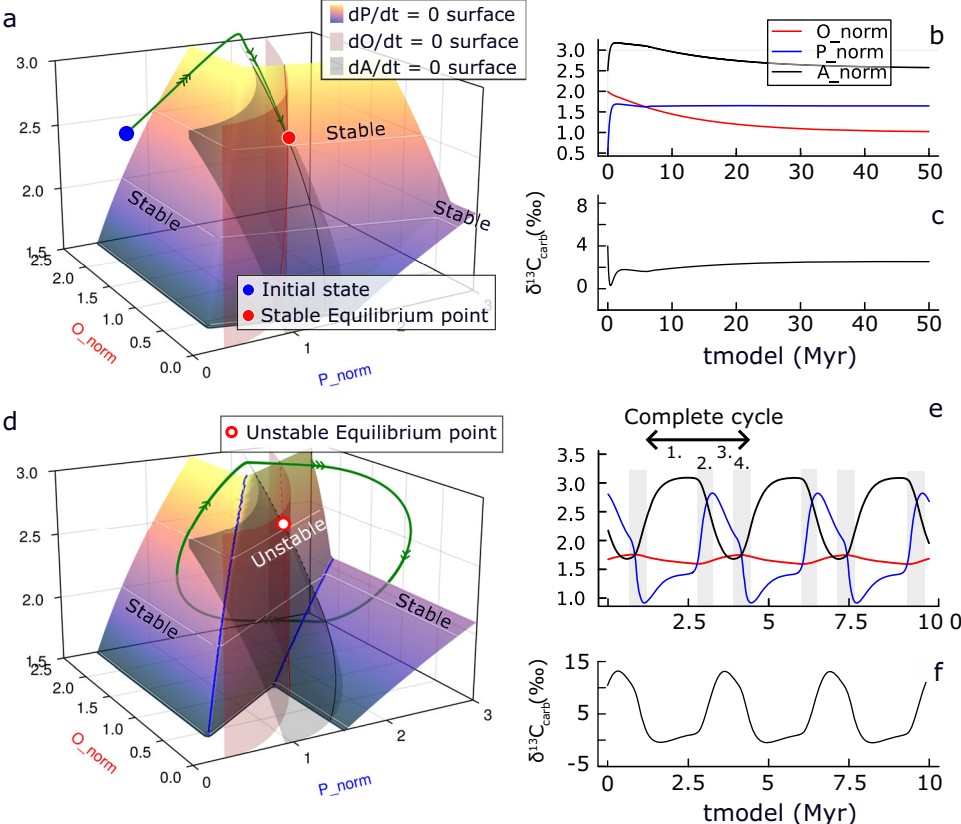

**Fig. 3 | Homogeneity of ocean floor redox conditions controls the stability of the phosphorus-carbon-oxygen system. a** The phase plane for an idealized stable system, showing the d$P$/d$t$ = 0 nullcline surface (plasma) where oceanic phosphorus level is invariant in time, and the d$O$/d$t$ = 0 (red) and d$A$/d$t$ = 0 (black) surfaces, and their intersections with the $P$ surface. **b, c** Time series of the normalized $\boldsymbol{P}$–$\boldsymbol{O}$–$\boldsymbol{A}$ levels and δ¹³C$_{carb}$. **d–f** The phase plane and time series for an

idealized "fast" $\boldsymbol{P}$–$\boldsymbol{O}$–$\boldsymbol{A}$ unstable system, showing limit cycle oscillations, omitting the adjustment process, with the $\boldsymbol{P}$ surface being folded with both unstable (repelling) and stable (attracting) regions. A complete cycle can be subdivided into four parts, the fast accumulation and consumption of the $\boldsymbol{P}$ are marked by gray bars in panel e. Note the different scales of the *tmodel* axis are given in various cases. X_norm represents the X level compared to the modern values.

The post-extinction flourishing of smaller phytoplankton therefore likely enhanced nutrient uptake efficiency ($k_u$), increasing the oxygen demand of bottom water as they sank in the early Triassic oceans. This is supported by previous modeling studies using box models[38,39], 3D-ocean-circulation models[56,57], or 1D box-diffusion models[58]. These show that doubling limiting nutrient uptake (e.g., increasing the nutrient uptake efficiency) or decreasing O₂ by half would result in global-wide ocean anoxia. As a result, the bottom waters of the early Triassic oceans were driven towards anoxia, and made more homogeneous in their redox state, thereby strengthening the anoxia-phosphorus-burial feedback and destabilizing the system (Fig. 2).

## Dynamical analysis reveals the source of unstable behavior
To illustrate how homogenizing the ocean floor redox condition controls the dynamics of the $\boldsymbol{P}$–$\boldsymbol{O}$–$\boldsymbol{A}$ cycles, we applied the excitable phosphorus oxygen carbon model (EPOC), which includes an idealized column-ocean domain[29] to account for the ocean's spatial redox heterogeneity (see Supplementary Information for details). The ocean is subdivided into 100 water columns: each of them has an independent redox state controlled by per-column parameterized efficiency of nutrient uptake ($k_u$)[36,39]. A new forcing ($S$) is introduced that acts as a multiplier to control the minimum value of $k_u$ in the water columns (while the maximum value of $k_u$ is fixed at 1.0). When $S$ is elevated, the range of $k_u$ values across 100 water columns shrinks, thereby increasing integrated oxygen demand across the water columns. Here we consider two idealized cases with $k_u$ ranging from 0.1 to 1.0 (stable) or from 0.7 to 1.0 (unstable), as shown in Fig. 3.

3D plots of the phase-planes (Fig. 3) are used to visualize the $\boldsymbol{P}$–$\boldsymbol{O}$–$\boldsymbol{A}$ differential equations (Table S3)[35,36,59]. For each variable, there is a surface ('nullcline') in 3D-phase space where it is invariant in time (d$P$/d$t$ = 0, d$A$/d$t$ = 0, and d$O$/d$t$ = 0). These three surfaces always have a unique intersection, which represents the equilibrium point of the system, which may be stable (red dot in Fig. 3a) or unstable (open red dot in Fig. 3d). The intersection lines of $P$ and $A$ surfaces, and $P$ and $O$ surfaces are marked by black and red lines, respectively. The evolution of the variables is expressed as a trajectory in the 3D-phase space (green line in Fig. 3a, d) or as a traditional time series (Fig. 3b, e).

If the marine burial flux is distributed over heterogenous redox conditions (represented by varying the efficiency of nutrient uptake $k_u$ from 0.1 to 1.0 across water columns), the strength of the globally integrated anoxia feedback is relatively small (Fig. 3a). Then a small increase in oceanic phosphorus results in a small increase in globally-integrated phosphorus burial (Fig. S7) and the system is stable. Starting from an arbitrary initialization, the normalized $\boldsymbol{P}$–$\boldsymbol{O}$–$\boldsymbol{A}$ reservoirs (Fig. 3b) and carbonate carbon isotope signature (Fig. 3c) find a unique steady state after a model adjustment time. The corresponding trajectory of variables $\boldsymbol{P}$–$\boldsymbol{O}$–$\boldsymbol{A}$ in phase space (Fig. 3a) shows this adjustment from an arbitrary initial point to the equilibrium point (an "attractor"). There is a relatively fast adjustment of $\boldsymbol{P}$ and $\boldsymbol{A}$ because their residence times (size of the reservoir (mol)/size of the flux (mol/y)) and corresponding response times are relatively short ($\boldsymbol{P}$ ~ 0.5 Myr and $\boldsymbol{A}$ ~ 1.0 Myr). This is followed by a slower adjustment of $\boldsymbol{O}$ (which has a longer response time ~5 Myr), along the black line defined by the intersection of the $\boldsymbol{P}$ and $\boldsymbol{A}$ surfaces. In this slow phase, $\boldsymbol{P}$ and $\boldsymbol{A}$ remain

close to source-sink balance, but the steady state of *A* decreases as the oxygen cycle slowly finds a balance at a lower *O* level.

Alternatively, if the marine burial flux is distributed over more homogenous redox conditions (with the efficiency of nutrient uptake $k_u$ varying from 0.7 to 1.0 across water columns), the anoxia feedback becomes stronger (Fig. 3d). Then a small increase in oceanic phosphorus causes a *decrease* in globally-integrated phosphorus burial as many water columns become anoxic at same time (Fig. S7, details in Supplementary Information), producing an unstable system. The criterion for instability in the *P*–*O*–*A* coupled system is that the *P* surface folds such that part of it becomes a "repelling" region, while the sides remain stable "attracting" regions (Fig. 3d). When the equilibrium point lies within this repelling region, it becomes an unstable "repeller" (open red dot, Fig. 3d). Small disturbances away from the equilibrium point (where $dP/dt = dA/dt = dO/dt = 0$) are magnified because at least one eigenvalue of the Jacobian for this repeller is positive[60]. Hence, the system trajectory no longer converges towards the equilibrium but instead moves around it. This corresponds to periodic solutions to the *P*–*O*–*A* differential equations.

In this idealized case (Fig. 3d), there are "fast" phosphorus-carbon self-sustaining (limit cycle) oscillations at almost constant oxygen level. The normalized *O* level is relatively high (in comparison to the modern value) with only modest fluctuations $O \sim 1.75 \pm 0.1$ (Fig. 3e). The oscillation is characterized by fast jumps in both marine phosphorus ($P \sim 2.0 \pm 1.0$) and atmosphere–ocean carbon ($A \sim 2.0 \pm 0.5$). A complete cycle spans ~3.2 Myr and can be divided into four stages (Fig. 3e). (1) Stage 1 (~1.6 Myr, 50% of the complete cycle): In the longest phase of the cycle the ocean is oxygenated with low phosphorus recycling efficiency (due to a low and redox-sensitive $C_{org}:P_{total}$ burial ratio[36,37]), low *P*, high *A*, and declining *O* due to low organic carbon burial. Elevated atmospheric $CO_2$ enhances silicate and phosphorus weathering, increasing the *P* reservoir. As *O* is also declining, this eventually triggers the onset of some anoxia in the bottom water. (2) Stage 2 (~0.5 Myr; 16% of the cycle): *P* and anoxia rapidly increase, supported by escalating phosphorus recycling from sediments (Fig. 2). This increases marine organic carbon burial, drawing down $CO_2$ (*A*) and slowly increasing oxygen (*O*). (3) Stage 3 (~0.6 Myr; 18% of the cycle): The drop in atmospheric $CO_2$, temperature, and phosphorus weathering causes *P* to shrink, while *O* is still rising, eventually triggering the return of oxic conditions to some bottom waters. (4) Stage 4 (~0.5 Myr; 16% of the cycle): The ocean rapidly oxygenates and *P* rapidly declines, supported by escalating phosphorus removal to sediments. As organic carbon burial plummets, *A* rises and *O* stabilizes, then slowly starts to decline (see Fig. S8 for the details of the cases in Fig. 3).

### Applying the model to the post-extinction interval

We now broaden the EPOC[29] to include the uranium cycle[61,62] and initialize it to represent an end-Permian steady state (details in "Materials and methods", and Fig. S9). We then use the model to explore what combination of factors can explain early Triassic oscillations in biogeochemical proxies.

Following previous studies[31], we adjusted value ranges of both Tectonic Uplift (*U*) and Vegetation (*V*) forcings (Fig. 2). *U* is elevated from its end-Permian value of 0.55[36] to 0.8, while *V* is reduced from 1.0 to 0.0 during 252–248 Ma and recovering linearly back to 0.25 at 246 Ma (Fig. 4a). We explore two scenarios for the forcing *S* that controls the minimum value of the efficiency of nutrient uptake ($k_u$) across the water columns. In the control model run, $S = 0.1 \pm 0.1$ remains constant, meaning the marine burial fluxes distribute over a heterogeneous seafloor redox condition for the entire interval of 254–244 Ma. In the treatment model run, we increase *S* to $0.4 \pm 0.1$ during the Induan-Olenekian (252–248 Ma) interval when microbialite depositions were widely distributed[53]. This raises the oxygen demand on the ocean floor in some locations by a factor of 3–5. See Eq. S10 and

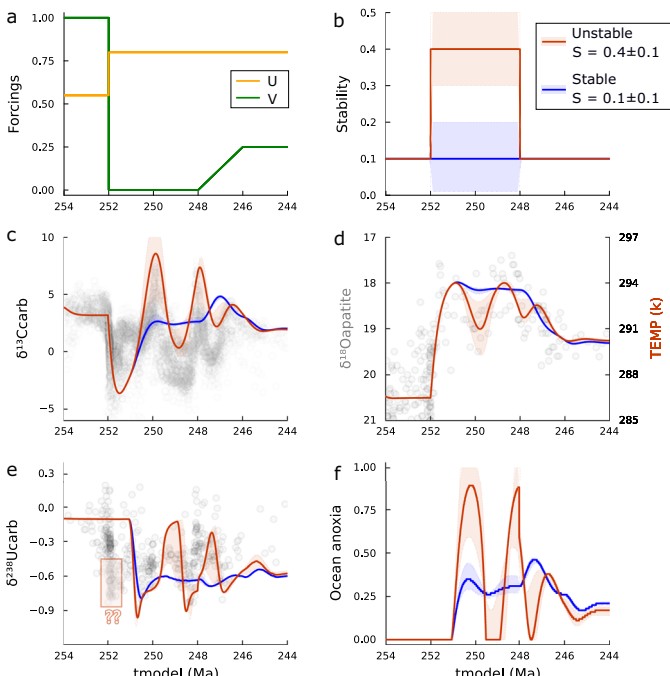

**Fig. 4 | Biogeochemical modeling results (time series). a** Tectonic Uplift (U) and Vegetation (V) forcings, representing the long-term effect of the volcanic eruption of the Siberian Traps and collapse of terrestrial ecosystem[31]. **b** System stability forcing (*S*), representing the degree of homogeneity of the ocean floor redox conditions. Two groups of model runs: lower *S* for heterogeneous ocean floor redox condition, hence a stable system (blue lines and bars), and higher *S* for homogenous ocean floor redox condition, hence an unstable system (orange lines and bars). **c**–**e** Comparison between modeled result and raw data of $\delta^{13}C_{carb}$, $\delta^{18}O_{apatite}$ vs temperature (*K*) and $\delta^{238}U_{carb}$, respectively. **f** Degree of marine anoxia. Note that the first negative uranium isotope excursion is not captured here, see extended model runs in Fig. S14, where the results are more consistent with data when short-term forcings[14] are considered.

the accompanying calculations, which show that reducing the cell size from the maximum values of the eukaryotic marine phytoplankton (e.g., 30 μm) to cyanobacteria (e.g., 10 μm)[63] leads to an increase in $k_u$ from 0.14 to 0.38. This homogenizes the water columns towards anoxia and destabilizes the system (Figs. S12 and S13). We now consider each scenario in turn. The sensitivity tests of the forcings VEG, S, and Uplift can be found in Figs. S10–S12, respectively.

### Post-extinction stable system

When the homogeneity of the ocean floor redox condition is set to remain stable over the P–Tr transition, the control model run shows that enhanced uplift and the loss of vegetation result in a greater flux of oxidative weathering (kerogen), which acts as both a carbon source and an oxygen sink (blue lines and bars in Fig. 4), as shown previously[31,32]. Moreover, the post-extinction shutdown of organic carbon burial on land, removes a carbon sink and oxygen source, driving the system to reach a new steady state, with higher atmospheric-ocean carbon (*A*) level and lower atmospheric-ocean oxygen (*O*) level. Concurrently, increased phosphorus weathering from silicate weathering leads to higher ocean phosphorus (*P*) levels at a Triassic steady state. These changes reproduce a ~1.5-Myr-duration negative $\delta^{13}C_{carb}$ excursion after the collapse of the terrestrial ecosystem (Fig. 4c), the early Triassic hothouse regime[8], due to high $pCO_2$ level (Fig. 4d), and an ocean anoxic event corresponding to the negative $\delta^{238}U_{carb}$ peak at ~251 Ma (Fig. 4e, f). The subsequent partial recovery of the terrestrial ecosystem produces cooling 248–246 Ma (Fig. 4d), but generates an erroneous peak in $\delta^{13}C_{carb}$

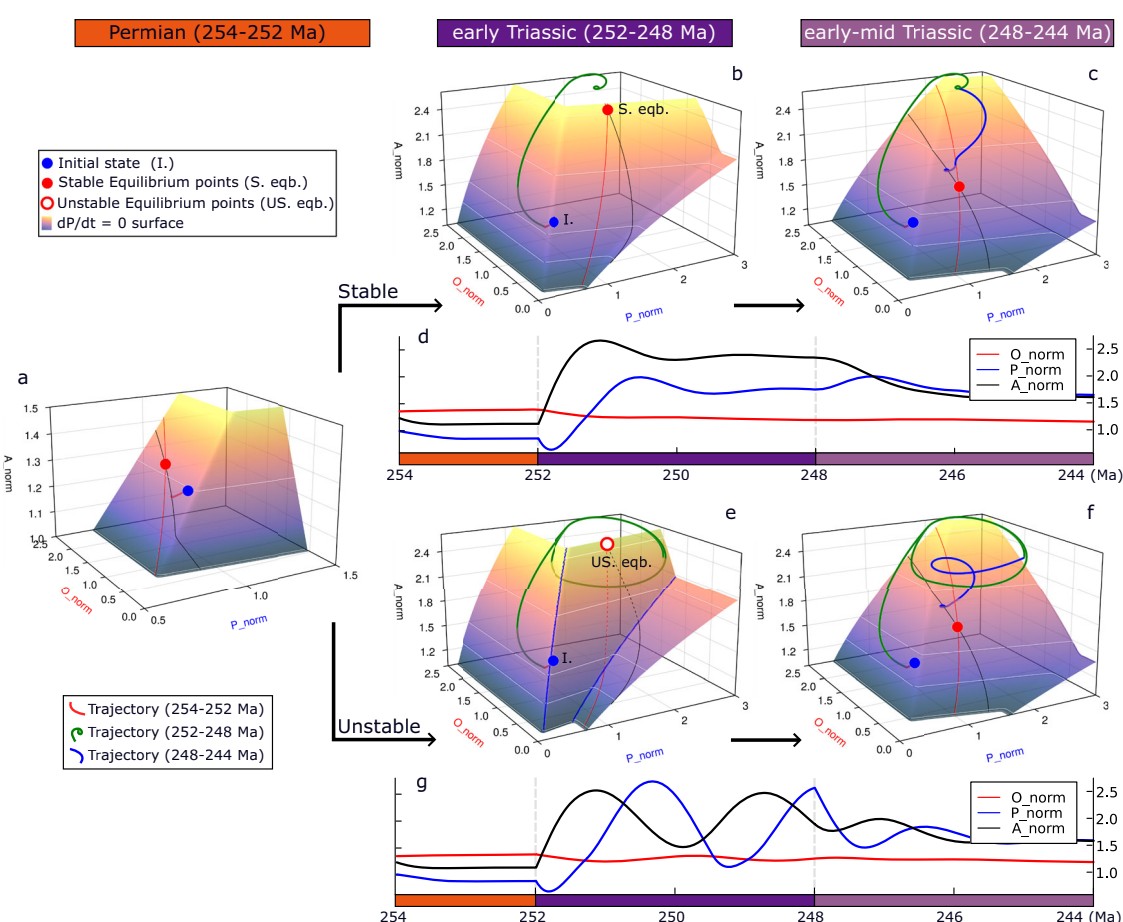

**Fig. 5 | Modeling results (phase planes) of biogeochemical cycles corresponding to the time series analyses (shown in Fig. 4).** Note that the oxygen and carbon nullcline surfaces are omitted, but the intersection lines (red and black lines on **P** surface) are shown. **a** Phase plane of the adjustment process during the end-Permian. Changing trajectories during 254–252 Ma (bold red line). **b, c** Phase planes of the control model run, changing trajectories during 252–248 Ma (bold green lines) and 248–244 Ma (bold blue lines). **d** Corresponding time series of reservoir **P**–**O**–**A** levels. **e, f, g** Phase planes of the treatment model run and corresponding time series of reservoir **P**–**O**–**A** levels. Note the control and treatment model runs both initialized from panel (**a**), and different scales in it are set to show the adjustment process from the end-Permian in detail. See the definition of the abbreviations in Fig. 3.

(Fig. 4c) (for sensitivity tests of the land plant recovery, see Figs. S10 and S11).

Importantly, the oscillatory variability in 252–244 Ma geochemical data cannot be captured. Allowing the forcing $S$ to increase from 0.0 to 0.2 during 252–248 Ma makes little difference, as the system remains stable, with marine burial fluxes continuing to be distributed across heterogeneous redox conditions.

**Post-extinction unstable system**

In the treatment model run where we modified the forcing $S$ to destabilize the system (Fig. 4b) the oscillations can be captured (orange lines in Fig. 4). The increase in bottom water oxygen demand and homogenization of the redox conditions of the water column (linked to the observed shift to smaller phytoplankton) produces oscillatory behavior in **P** and **A** (and to a lesser degree **O**) (Fig. 5). Model predicted $\delta^{13}C_{carb}$ matches up with the dramatic and periodic oscillations, including the great positive excursions at -250 Ma, 248 Ma, and 246.5 Ma (and the negative excursions in between), and the gradually diminishing oscillation amplitude (Fig. 4c). Temperature oscillations within the hothouse condition are captured (Fig. 4d). Meanwhile, the unstable system produces oscillations between a fully anoxic and an oxygenated ocean state (rather than persistent partial anoxia) that broadly match the $\delta^{238}U_{carb}$ variability including three negative excursions at -251 Ma, 249 Ma, 247 Ma, respectively (Fig. 4e, f).

The exception is that the very first negative $\delta^{238}U_{carb}$ excursion (252 Ma) is not simulated in Fig. 4, as the model runs do not account for short-term forcings during the Permian–Triassic transition (-0.5 Myr scale). Notably, previously proposed short-term forcings, such as light carbon input, extra phosphorus input, and oxidation of terrestrial biomass[14,20,31,64] can augment our forced-stability-response view. Extended model runs incorporating these short-term forcings cannot directly generate the oscillations in a stable system. However, they successfully reproduce both the earliest Triassic excursions and the early Triassic oscillations in an unstable system. This further supports our hypothesis that system stability, rather than short-term forcings, is the primary driver of the oscillatory behavior observed during the early Triassic. (Supplementary Information Section 5, Fig. S14).

A phase-plane analysis illustrates the changing dynamical behavior of the system from the end-Permian (254–252 Ma) to the early Triassic (252–248 Ma) and early-middle Triassic (248–244 Ma) (Fig. 5). Driven by the changes of forcings $U$ and $V$, the equilibrium point of the system (red point in phase planes) is moved twice, and the system adjusts to track the new equilibrium point. Owing to the relatively fast cycles of **P** (-0.5 Myr) and **A** (-1.0 Myr), in both the control and treatment model runs, the system moves quickly towards the intersection line between **P** and **A** surface (black line) during the end-Permian to middle Triassic, but the oxygen cycle (-5 Myr) does not reach source-sink balance. The major difference in the treatment model run is that

the stable equilibrium point is turned into an unstable "repeller" during the early Triassic and the system exhibits self-sustaining (or limit cycle) periodic (2–3 Myr) oscillations (Fig. 3). These follow the orbiting trajectory around the unstable equilibrium point (open dot, Fig. 5e) in the 3D-phase space. Unlike the forced-response view where the changes in $U$, $V$ forcings only change the magnitude of the response, here changes in the stability forcing $S$ both change the magnitude of the response and cause it to fluctuate periodically (Figs. 4 and 5).

### Unstable behavior dominated the early Triassic biogeochemical cycles

New modeling shows that the periodic oscillations (in $\delta^{13}C_{carb}$, $\delta^{238}U_{carb}$, and $\delta^{18}O_{apatite}$) observed after the EPME were not triggered by multiple episodes of light carbon input from different sources, instead, the oscillatory behavior was due to destabilization of the coupled phosphorus, carbon, and oxygen biogeochemical cycles. Emplacement of the Siberian Traps accompanied by the collapse of land ecosystems after the EPME can explain a first negative excursion in $\delta^{13}C_{carb}$ and subsequent recovery, persistent hothouse conditions in the early Triassic, and a single persistent negative excursion in $\delta^{238}U_{carb}$. However, to explain observed oscillations, we have to invoke an increase in nutrient uptake efficiency in the ocean and associated homogenization of ocean floor redox conditions. This was plausibly linked to the collapse of the size-structure marine ecosystem, leaving an ocean dominated by small phytoplankton cells, which exhibit high nutrient uptake efficiency. This increased oxygen demand tends to homogenize the ocean floor redox condition towards anoxia and produces a stronger anoxia-dependent feedback on phosphorus recycling after the EPME. This is the key driver forcing system instability in our model (Figs. 3 and 4). Persistent hothouse conditions and the recurring expansion of ocean anoxia plausibly delayed the recovery of marine ecosystems. Although the EPME is recognized as the most severe mass extinction event in the Phanerozoic, its impact on Earth system stability seems to have been underestimated.

## Methods

A total of 7491 $\delta^{13}C_{carb}$ data, 468 $\delta^{238}U_{carb}$ data, and 616 $\delta^{18}O_{apatite}$ data are compiled here and available in Supplementary Information. The age model for geochemical profiles is based on conodont-biostratigraphic and carbon isotope-stratigraphic frameworks[65].

### Biogeochemical model

The model employed is an extended version of the EPOC established by ref. 29, that adds a two-sink uranium cycle module to predict the variation of $\delta^{238}U_{carb}$ values[61,62]. The model captures the long-timescale ($10^5$–$10^8$ years) evolution of ocean phosphorus (P), global atmosphere–ocean oxygen (O), and carbon (A) controlled by weathering and burial fluxes. The EPOC model includes a minimal representation of land surface carbonate and silicate weathering, oxidative weathering of organic carbon, and phosphorus weathering (see equations in Table S3), using parameterizations from the GEOCARB and COPSE models[35,36,59]. Marine burial fluxes for organic carbon and redox-sensitive phosphorus are represented by an idealized ocean column model that parameterizes the spatial distribution of these burial fluxes across a heterogeneous ocean floor redox state[29]. Phosphorus is treated as the ultimate limiting nutrient for ocean productivity on geological time scales ($>10^6$ years), since any short-term nitrogen limitations of primary productivity are balanced by N-fixation and denitrification fluxes[66]. The new model configurations are developed using PALEOtoolkit (https://github.com/PALEOtoolkit). The detailed model description is available in the Supplementary Information.

### End-Permian steady state

The end-Permian steady state (Fig. S9) has ocean phosphorus $\cong 1$ of present ocean level (2.15 µmol/kg[36]), $pO_2$ (atm) $\cong 0.28$[67], $pCO_2$

(PAL) $\cong 1.5$ (420 ppmv, compare to pre-Industrial Revolution value of 280 ppmv[68]). This is achieved by fixing the physical forcings inherited from the GEOCARB and COPSE models[35,36,59] at values for the end-Permian (252 Ma) (Table S4). To obtain the ~4‰ background $\delta^{13}C_{carb}$ signal, we assume $\delta^{13}C_{carb}$ values of 1.0‰ and −22‰ for sedimentary rock reservoirs of organic (reduced) carbon and carbonate (oxidized) carbon, respectively.

### Sensitivity tests and extended model runs

The sensitivity tests of the forcing VEG (different recovery patterns), S (different patterns), and Uplift (different values and decay scenarios) are shown in Figs. S10–S12, respectively. The extended model runs, based on the configuration in Fig. 4, incorporate short-term forcings across the PTB (e.g., enhanced oxidative weathering, light carbon input, and additional phosphorus input in Figs. S13 and S14) to reproduce both the short-term isotopic excursion around 252 Ma and the long-term oscillations of the early Triassic. These sensitivity tests and extended runs collectively reinforce the conclusion that system stability, rather than short-term forcings, governs the oscillatory behavior observed in the early Triassic.

### Alternative dynamical hypotheses and model limitations

The simplest hypothesis of nonlinear dynamics (explored here) is a transition to an unstable limit-cycle-oscillation regime. However, given the available data, we cannot exclude a transition to a marginally-stable regime (with lightly damped oscillations), or to an excitable, marginally-stable regime with small (hence as yet undetected in the geochemical record) stochastic forcing of quasi-periodic limit-cycle excursions[29]; however, all these hypotheses include a carbon-phosphorus oscillation or limit-cycle as the central mechanism.

The structure of the ocean domain (the "column ocean") used here is necessarily simple, more realistic paleo-geography is not considered here. Given the paleo-geography of the early Triassic, it is plausible that restricted water exchange between Neo-, Paleo-Tethys, and other ocean basins may play a role in driving oscillation of the biogeochemical cycles[69].

Small phytoplankton cells not only enhance nutrient uptake efficiency, as demonstrated in this study, but also reduce the export efficiency of organic matter to deeper waters (>100 m). This has the potential to counteract increased oxygen demand within the water column while simultaneously decreasing the overall burial flux of organic carbon, thereby reducing the oxygen source. Consequently, explicit consideration of the biological pump in the "column ocean" is needed in the future to better understand the relationship between cell size and the stability of phosphorus-dominated biogeochemical cycles.

## Data availability

The data that support the findings of this study (Fig. 1) are available from the Supplementary dataset or Z.L. (zihengli@cug.edu.cn) on request.

## Code availability

The code for the EPOC model is available from Z.L. (zihengli@cug.edu.cn) or S.J.D. (S.Daines@exeter.ac.uk) on request. Or the release version on Github (https://github.com/PALEOtoolkit).

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

## Acknowledgements

This study was funded by a grant (41930322) of the NSFC (Z.Q.C.) and a grant (42488201) of the Basic Science Center Program. Z.L. was supported by the programs of China Scholarships Council (no. 202106410094). Both S.J.D. and T.M.L. were supported by the Natural Environment Research Council (grant NE/T008458/1). F.-F.Z. was supported by the National Natural Science Foundation of China (Grant No. 42261144668 and Grant No. 42293280).

## Author contributions

Z.L. conceived the project, performed model runs, analyzed results, and wrote the first draft of the paper. Z.-Q.C., T.M.L., and F.F.Z. supervised the project. S.J.D. and Z.L. created the model. All authors contributed to the revision of the manuscript.

## Competing interests

The authors declare no competing interests.
