## [Transparent Peer Review file · Nature Communications]

Earth system instability amplified biogeochemical oscillations following the end-Permian mass extinction

Corresponding Author: Professor Zhong-Qiang Chen

Version 0:

Reviewer comments:

Reviewer #1

(Remarks to the Author)

The authors present a new box model to investigate regular oscillations in ocean biogeochemistry that occurred over the early Triassic following the end-Permian mass extinction. The authors show convincingly that periodic changes in carbon and uranium isotopes recorded over this period can be explained by internal feedbacks in the coupled carbon-phosphorus-oxygen system. They also discuss that periodic external forcing via e.g. volcanic degassing is not supported by the geological record. They, hence, conclude that the regular and periodic changes in environmental conditions that occurred over the early Triassic reflect the internal instability and nonlinear dynamics of the biogeochemical system rather than external forcing. I fully support this conclusion and the publication of their manuscript in Nature Communications. The manuscript is well written but I have a few critical comments to the box model used by the authors.

1. The authors assume that ocean productivity is ultimately controlled by phosphate on geological time scales. However, plankton productivity is limited by N and Fe rather than P over most of the modern ocean. Moreover, a decline in ocean oxygenation may induce a shift from Fe/P to N limitation and a biogeochemical amplification of deoxygenation via benthic P and Fe release is only possible if N-fixation is strongly amplified to compensates for N losses induced by denitrification (Wallmann et al., 2022). Since we do not even understand the controls on nitrogen fixation in the modern ocean, we cannot know whether rates of nitrogen fixation were high enough in the geological past to overcome N limitation. Nevertheless, the authors argue that ocean productivity is limited by P rather than N over the early Triassic. Moreover, they ignore Fe limitation in their model even though it may affect phytoplankton growth over the model period. A recent modeling paper actually showed that periodic oscillations can also occur in a global ocean model that considers changes in N, P and Fe inventories and their effects on ocean productivity where the redox-dependent Fe cycle plays a key role (Wallmann et al., 2019). The authors should, hence, add a paragraph to their manuscript where they discuss the role of N, P, and Fe limitation in the modern ocean and the potential role of these limiting nutrient in the early Triassic ocean. In this section, they should also discuss how their assumption that P is the only limiting nutrient may affect the model results.

2. The global ocean model used by the authors seems to employ just one water box representing the global ocean. This box is coupled to a heterogenous seafloor to simulate redox gradients across the seabed. Internal oscillations presented in the model only occur when the seabed exhibits a small redox variability ($S = 0.4$) and disappear when a high variability is applied ($S = 0.1$). It is unclear whether this conclusion would also hold when multiple ocean boxes were used in the model. A previous multi-box model of the Cretaceous ocean actually showed that oscillations only occur when the lateral water exchange between a key ocean box (proto North Atlantic) and other ocean boxes is restricted (Wallmann et al., 2019). Given the paleo-geography of the late Triassic, it seems to be plausible that the Tethys and its restricted water exchange with other ocean basins may play a similar key role in biogeochemical cycling (Jurikova et al., 2020). The authors should, hence, emphasize that the low redox variability is only a necessary condition for the development of periodic ocean changes when a one-box model is applied and that higher redox variability may also allow for these internal oscillations when multiple ocean boxes are used.

3. The authors argue that a shift to smaller phytoplankton induces an increase in oxygen demand in bottom waters that limits the redox variability at the seabed ($S = 0.4$) and allows for oscillatory behavior. However, most of the export production and rain of particulate organic matter to the seabed is driven by large rather than small plankton in the modern ocean. A shift to smaller plankton may, hence, decrease the oxygen demand in bottom waters and increase rather than decrease the redox variability at the seabed. The authors also state that small phytoplankton has a higher nutrient uptake efficiency than large

plankton. It is, however, more likely that P uptake efficiency increases when other limiting nutrients (N, Fe) are available such that P can be consumed by phytoplankton. The authors' hypothesis that a shift to smaller phytoplankton is responsible for the oscillatory behavior might be valid within the framework of their simple model but is very likely not valid for the much more complex natural system. It seems to be more likely that a high availability of N and Fe is needed to promote P uptake efficiency and oscillatory behavior.

References

Jurikova, H., Gutjahr, M., Wallmann, K., Flogel, S., Liebetrau, V., Posenato, R., Angiolini, L., Garbelli, C., Brand, U., Wiedenbeck, M., & Eisenhauer, A. (2020, Nov). Permian-Triassic mass extinction pulses driven by major marine carbon cycle perturbations [Article]. *Nature Geoscience*, 13(11), 745-+. <https://doi.org/10.1038/s41561-020-00646-4>

Wallmann, K., Flogel, S., Scholz, F., Dale, A. W., Kemena, T. P., Steinig, S., & Kuhnt, W. (2019, Jun). Periodic changes in the Cretaceous ocean and climate caused by marine redox see-saw [Article]. *Nature Geoscience*, 12(6), 456- 462. <https://doi.org/10.1038/s41561-019-0359-x>

Wallmann, K., Jose, Y. S., Hopwood, M. J., Somes, C. J., Dale, A. W., Scholz, F., Achterberg, E. P., & Oschlies, A. (2022, May). Biogeochemical feedbacks may amplify ongoing and future ocean deoxygenation: a case study from the Peruvian oxygen minimum zone. *Biogeochemistry*, 159(1), 45-67. <https://doi.org/10.1007/s10533-022-00908-w>

(Remarks on code availability)

Reviewer #2

(Remarks to the Author)

The authors employed an ocean phosphorus (P), global atmosphere-ocean oxygen (O), and carbon (A) -coupled biogeochemical model to explore the periodic oscillations in isotopic signatures during the 10-million-year interval of the Early Triassic following the Permian-Triassic mass extinction. Through modeling and sensitivity analysis, these oscillations are attributed to the destabilization of coupled phosphorus, carbon, and oxygen biogeochemical cycles, driven by the homogenization of ocean floor redox conditions caused by an increase in nutrient uptake efficiency within the ocean. Overall, this is a concise manuscript. However, I am concerned that some critical evidence supporting the authors' conclusions is missing, and additional explanations are required to fully substantiate their claims and ensure for publication.

Major comments

1. The correlation with geological records remains unclear (Figure 4). While I acknowledge that the biogeochemical model indicates an unstable system resulting in periodic oscillations in carbonate carbon and uranium isotopes during the Early Triassic, the model outputs appear to deviate from geological records, particularly regarding uranium isotopes (Figure 4E, showing a time shift between the records and model outputs) and temperature (Figure 4D, with mismatched magnitudes at both the maximum around 252 Ma and the minimum around ~247 Ma). This discrepancy hinders confidence in determining whether adjustments to the uptake efficiency parameters are valid.

Additionally, I recommend avoiding the use of light-grey color for the geological records in the figure. Clear visual distinctions between the model outputs and geological records are essential for effective comparison.

2. The instability of the system appears to arise from homogeneous redox conditions, with the limited range of uptake efficiency among small-celled phytoplankton (e.g., cyanobacteria) across the ocean identified as a key factor. However, I question whether convincing evidence exist to support this fundamental assumption.

Cyanobacteria are typically abundant in oligotrophic marine environments, where they thrive due to smaller cell sizes and highly efficient nutrient uptake systems. This enables them to outcompete other phytoplankton. However, when nutrient levels increase and become abundant, cyanobacteria may lose this competitive advantage, and eukaryotic algae generally outcompete cyanobacteria due to their larger size, higher nutrient demands, and faster growth rates. This consensus appears to contradict the authors' argument, which suggests that enhanced nutrient input, driven by increased weathering, promotes the proliferation of small-celled phytoplankton (e.g. cyanobacteria).

Additionally, regarding the parameter S in the model simulation, are there any direct geological records or indirect evidence (e.g., statistically supported increases in burial) to support $S=0.4\pm 0.1$ instead of $S=0.1\pm 0.1$ during the interval between 252 Ma and 248 Ma? This distinction is crucial, as parameter adjustments should be grounded in robust and valid evidence.

3. In the model simulation, the uplift external forcing remains at a high level after 252 Ma. I wonder whether this timing is well-aligned with the Siberian Traps. According to Burgess et al. (2017), the maximum age for the cessation of Siberian Traps activity is estimated at 250.2 ± 0.3 Ma. Beyond this point, it would be challenging to sustain intensified silicate weathering and the associated enhanced input of phosphorus to the oceans. Have you conducted a sensitivity analysis of the uplift forcing (U) to address this temporal discrepancy?

4. Definitions of stable and unstable. Mathematically, based on lines 248–250, I can understand how the inflection point occurs as the range of ku_{\min} is adjusted. However, it remains difficult to quantitatively visualize the transitional conditions under which the system becomes unstable. For instance, would it be possible to include a figure illustrating the distribution of homogeneous redox conditions (e.g., anoxia extent) across 100 ocean columns in the model? Such a visualization could

provide clearer insights into the transition to instability.

Minor Comments

Lines 124–127: The argument appears self-contradictory. Are you suggesting that silicate weathering has a significant impact, whereas the other two factors have less impact? Please clarify this point.

Line 161: Could you elaborate on the meaning of "global marine burial phosphorus burial flux"? The phrasing is somewhat unclear.

Line 203: There is a typo: "He we consider..." should be corrected.

Lines 306–309: Are you referring to Figure 4 rather than Figure 3? Please verify.

(Remarks on code availability)

Version 1:

Reviewer comments:

Reviewer #1

(Remarks to the Author)

The manuscript has been greatly improved and the response letter of the authors addresses most of the reviewer's comments. However, I still have a few comments and questions that the authors may want to address:

I agree that small phytoplankton uses nutrients more efficiently than large phytoplankton. However, large phytoplankton usually sinks more rapidly than small phytoplankton. It is, thus, likely that the export efficiency decreases when the plankton ecosystem is dominated by small plankton. This effect could lead to a smaller oxygen demand at > 100 m water depth that could partially counteract the increase in oxygen demand induced by enhanced nutrient utilization in the surface ocean. The authors may want to discuss these effects of plankton size on export efficiency and oxygen demand in section "Alternative dynamical hypotheses and model limitations".

The nitrogen cycle included in the extended model assumes that nitrogen fixation compensates for denitrification. It is not clear whether this assumption is actually valid. This is an important caveat since negative feedbacks inherent to the nitrogen cycle would strongly affect the system behavior if nitrogen fixation could not compensate for enhanced denitrification during anoxic phases. The model assumption applied by the authors also implies that nitrogen fixation rates were greatly enhanced during periods of ocean anoxia to compensate for enhanced denitrification. Is there any evidence in the geological record that supports this model outcome? Are biomarkers for N-fixing phytoplankton and/or microbialites/MISS showing higher concentrations/abundances during anoxic compared to oxic phases? Such data would greatly support the model hypothesis. If these data are not available, the authors may want to discuss this lack of evidence and the possible system effects of reactive nitrogen loss during anoxia in section "Alternative dynamical hypotheses and model limitations".

(Remarks on code availability)

Reviewer #2

(Remarks to the Author)

The authors have satisfactorily addressed most of my comments, particularly regarding the geological record support and the sensitivity analysis of uplift forcing. I fully endorse publication of the manuscript in Nature Communications. The clarifications provided in response to my remarks strengthen the paper and will benefit readers. However, I recommend adding more context—at least in the supplementary materials—on how oceanic nutrient levels influence primary production and contribute to anoxic conditions. This additional information would further enhance the manuscript's clarity and robustness.

(Remarks on code availability)

Reviewer #1 (Remarks to the Author):

The authors present a new box model to investigate regular oscillations in ocean biogeochemistry that occurred over the early Triassic following the end-Permian mass extinction. The authors show convincingly that periodic changes in carbon and uranium isotopes recorded over this period can be explained by internal feedbacks in the coupled carbon-phosphorus-oxygen system. They also discuss that periodic external forcing via e.g. volcanic degassing is not supported by the geological record. They, hence, conclude that the regular and periodic changes in environmental conditions that occurred over the early Triassic reflect the internal instability and nonlinear dynamics of the biogeochemical system rather than external forcing. **I fully support this conclusion and the publication of their manuscript in Nature Communications.** The manuscript is well written but I have a few critical comments to the box model used by the authors.

REPLY: We sincerely thank you for your constructive suggestions and critical comments, which significantly improved the quality of our manuscript.

The authors assume that ocean productivity is ultimately controlled by phosphate on geological time scales. **However, plankton productivity is limited by N and Fe rather than P over most of the modern ocean.** Moreover, a decline in ocean oxygenation may induce a shift from Fe/P to N limitation and a biogeochemical amplification of deoxygenation via benthic P and Fe release is only possible if **N-fixation is strongly amplified to compensates for N losses induced by denitrification** (Wallmann et al., 2022).

REPLY: Yes, N is often the proximately limiting nutrient, but it tends to track longer timescale variations in P. We now explicitly add N cycling to our previous model to show that this does not affect the predictions. The periodicity and dynamics of the ~1-Myr to 20-Myr geological limit cycle are controlled by the timescale-separation

between P-O-A (plus N) geochemical cycles. Where fast variables (e.g., P and N cycles) quickly reach equilibrium at short timescales, maintaining the slow variables (e.g., O and A cycles) in a quasi-steady state until the fast variables undergo rapid changes again, this is the essence of the periodic solutions of the P-O-A plus N differential equations (details in Fig. 3).

The N cycle ($\cong 4.35e16$ mol) in the ocean is balanced by nitrogen fixation (as the source, $\cong 8.75e12$ mol/yr), and denitrification and anammox (as the sink, $\cong 8.6e12$ mol/yr). This gives N a short residence time in the ocean ($\tau_N = 4.35e16$ mol / $8.75e12$ mol/yr $\cong 0.5e4$ yr), implying that nitrogen fixation can replenish the ocean nitrogen supply to satisfy biological requirements at the timescale of $0.5e4$ yr. Therefore, when we consider periodic oscillations at the geological timescale ($\tau_P \cong 1e5$, $\tau_A \cong 1e6$, $\tau_O \cong 5e5$), the N-cycle is always balanced (in steady state).

If the P content of the ocean changes, the N content adjusts through its source-sink balance. In this way, the concentration of P in the oceans becomes the ultimate limiting nutrient for primary production. This is the geochemical-view of the oceanic nutrient-limitation (Broecker and Peng, 1982; Van Cappellen and Ingall, 1997; Bjerrum and Canfield, 2002; Bergman et al., 2004; Reinhard et al., 2016; Lenton et al., 2018; Canfield et al., 2020).

Since we do not even understand the controls on nitrogen fixation in the modern ocean, we cannot know **whether rates of nitrogen fixation were high enough in the geological past to overcome N limitation**. Nevertheless, the authors argue that ocean productivity is limited by P rather than N over the early Triassic.

REPLY: Nitrogen fixation does not need to overcome N limitation, it just needs to respond to variations in P so that N tracks variations in P.

Considering this factor, we added a new model configuration with enabling N cycle. To test the view that “nitrogen fixation was not high enough to overcome the N limitation”, we use the following equations (Bergman et al., 2004; Lenton et al., 2018) to construct the N cycle, in which **nitrogen fixation is never enough to overcome N limitation**:

$$\text{newp} = r_{C:P} * \min(30.9 * \frac{\mathbf{n}}{r_{N:P}}, 2.2 * \mathbf{p})$$

where the new production (newp) is controlled by both nutrient phosphate and nitrate potentially limiting. The values 30.9 and 2.2 are the present average concentrations of nitrate and phosphate in the ocean (in $\mu\text{mol/kg}$). Both $r_{C:P} = 117$ and $r_{N:P} = 16$ are the Redfield ratios, and N therefore is proximately limiting at present giving $\text{newp}_0 = 225.96$ as a normalized constant. The \mathbf{n} and \mathbf{p} are the nitrate and phosphate reservoir size normalized to the present level.

$$\text{denit} = k_{\text{denit}} * \left(1 + \frac{\text{anox}}{1 - k_{\text{oxic}}}\right) * \mathbf{n}$$

where the denitrification (denit) increases as anoxia increases. The parameters k_{denit} ($4.3\text{e}12 \text{ mol/yr}$) and $k_{\text{oxic}} (=0.86)$ are the denitrification constant and the present oxic fraction, respectively. The anox is the anoxic fraction of the ocean.

$$\text{nfix} = k_{\text{nfix}} * \left(\frac{\mathbf{P} - \frac{\mathbf{N}}{r_{N:P}}}{\mathbf{P}_0 - \frac{\mathbf{N}_0}{r_{N:P}}}\right)^2 \text{ for } \frac{\mathbf{N}}{r_{N:P}} < \mathbf{P}, \text{ else } 0$$

where the nitrogen fixation (nfix) keeps the nitrate below its Redfield ratio with phosphate and is never be enough to overcome the N limitation. The uppercase bold N and P are the reservoirs sizes in mol, both \mathbf{N}_0 ($4.35\text{e}16$) and \mathbf{P}_0 ($3.1\text{e}15$) are the present-day reservoirs sizes. Marine organic nitrogen burial is a small term dependent on marine organic carbon burial and fix C:N burial ratio = 37.5 (Lenton et al., 2018): $\text{monb} = \text{mocb}/\text{CN}_{\text{sea}}$.

Figure S15. (A-B) The variation of reservoirs P and N, (C-D) their sources and sinks, and (E-F) the carbon and uranium isotopic results. The forcings are set constant to run the test in an unstable system: VEG = 0.0, S = 0.4, Uplift = 0.8.

Figure S15 shows that adding the N cycle does not fundamentally alter the P-O-A-dominated dynamic system, once the physical forcings are set close to the major model run in the manuscript, the ~ 1.5 Myr periodic oscillation can be observed.

Importantly, during the limit cycle oscillation, the P cycle is characterized by source-sink imbalance, leading to sharp changes in oceanic P levels and control the dynamics of the system, while the N cycle remains in source-sink balance (or steady state) throughout the whole oscillation (Figure S15 C-D).

These results agree well with the typical geochemical-view that although the N has always been the limiting nutrient of the primary production ($N:P < 16$) in our model run, the N cycle reaches equilibrium at $0.5e4$ (yr) timescale with reasonably high nitrogen fixation flux, implying that the N cycle can potentially have great impact on the short-timescale perturbations e.g., the geochemical anomalies during the onset of the very first carbon isotope excursion (~ 251.9 to 251.5 Ma). But on the geological timescale (~ 1 to 20 Myr), the N cycle can be seen as being in equilibrium throughout the entire Early Triassic, and P becomes the ultimate limiting nutrient and dominates the periodic oscillations of the system.

Moreover, they ignore Fe limitation in their model even though it may affect phytoplankton growth over the model period. A recent modeling paper actually showed that periodic oscillations can also occur in a global ocean model that considers changes in N, P and Fe inventories and their effects on ocean productivity where the **redox-dependent Fe cycle plays a key role** (Wallmann et al., 2019).

REPLY: We apologize for omission of such an important reference (Wallmann et al., 2019). This reference combining with our results reveals the most important nature of the periodic oscillations across different geological intervals, which is the *period* itself. The N-P-Fe dynamic system (or differential equations) contains limit cycle regimes with **~40 kyr** period, in which N and Fe are the fast variables, leaving P as the slow variable. In this case, the upper limit for the period of the limit cycle is set by the residence time of P τ_P (1e5 years). Therefore, the N-P-Fe system alone cannot explain the Early Triassic oscillations with an **~1.5-Myr** period, which is **two orders of magnitude greater** than the period in Wallmann et al. (2019). Instead, we must invoke O and A as slow variables.

Independently, previous studies suggested that Fe limitation was more likely to have occurred in the Late Permian, rather than in the Early Triassic (Sun, 2024 <https://doi.org/10.1016/j.earscirev.2024.104914>). Aeolian dust and riverine influx, boosted by arid climates and monsoons, along with volcanic eruptions like the Siberian Traps, provided significant Fe. Anoxic conditions, covering large part of the seafloor, facilitated Fe remobilization via an anoxic Fe shuttle, ensuring long-distance transport to pelagic regions. The expansion of anoxia enabled diverse Fe cycling scenarios, maintaining availability despite localized pyrite burial. These mechanisms ensured sufficient Fe supply, making Fe limitation improbable except under specific conditions, such as oxygenated waters with restricted aerosol transport and pre-existing N and P limitations.

The authors should, hence, **add a paragraph to their manuscript where they discuss the role of N, P, and Fe limitation** in the modern ocean and the potential role of these limiting nutrient in the early Triassic ocean. In this section, they should also discuss how their assumption

that P is the only limiting nutrient may affect the model results.

REPLY: Thanks, we've added a new paragraph to discuss the limitations of primary production, and emphasise the critical influence of the N-Fe cycles on the short-timescale geochemical perturbations after the PTB.

2. The global ocean model used by the authors seems to **employ just one water box representing the global ocean**. This box is coupled to a heterogenous seafloor to simulate redox gradients across the seabed. Internal oscillations presented in the model only occur when the seabed exhibits a small redox variability ($S = 0.4$) and disappear when a high variability is applied ($S = 0.1$). **It is unclear whether this conclusion would also hold when multiple ocean boxes were used in the model.**

REPLY: We do not use a single box ocean model, instead the ocean domain consists of a set of columns ($n = 100$). Water column oxygen demand varies between columns controlled by per-column parameters k_{U_i} (the nutrient efficiency of nutrient uptake, Canfield, 1998; Lenton et al., 2018). Conceptually, we order the columns by increasing nutrient uptake efficiency (see details in Section 3.1 in the SI).

Daines and Li (2024) explicitly considered a hierarchy of two models and compared an intermediate-complexity ocean model (Romaniello and Derry, 2010) to the column model used here, and showed how the column model is a consistent representation of the behaviour of the multi-box model on the timescales considered here (long relative to the ocean circulation timescale of $\sim 1e3$ yr).

In essence, the limit cycle oscillation represents the periodic solution of the P-O-A differential equations, which is independent to the complexity of the ocean domain. The necessary condition for the appearance of the limit cycle is the marine integrated P burial becomes a non-monotonic function of the oceanic P level (Fig S7) when the P and organic carbon burial in homogeneous ocean floor redox condition (see details in Section 3.2 in the SI).

Figure S7. Local redox dependence combines with globally averaged oceanfloor oxygen to determine global marine phosphorus burial. (A) Per-column oceanfloor oxygen demand kU_i , $ncol = 100$, $(kU_{min}, kU_{max}) = (0.1, 1.0)$ and $(0.7, 1.0)$ for stable and unstable system, respectively. (B) Fraction of global marine organic carbon burial under anoxic conditions as a function of normalized marine phosphorus level at constant atmosphere-ocean oxygen and carbon. (C) Global marine phosphorus burial as a function of normalized marine phosphorus at constant atmosphere-ocean oxygen and carbon.

A previous multi-box model of the Cretaceous ocean actually showed that **oscillations only occur when the lateral water exchange between a key ocean box (proto North Atlantic) and other ocean boxes is restricted** (Wallmann et al., 2019). Given the paleo-geography of the late Triassic, it seems to be plausible that the **Tethys and its restricted water** exchange with other ocean basins may play a similar key role in biogeochemical cycling (Jurikova et al., 2020). The authors should, hence, emphasize that the **low redox variability is only a necessary condition** for the development of periodic ocean changes when a one-box model is applied and that higher redox variability may also

allow for these internal oscillations when multiple ocean boxes are used.

REPLY: We've added the heuristic discussion to main text following your suggestions (See line XXX).

3. The authors argue that a shift to smaller phytoplankton induces an increase in oxygen demand in bottom waters that limits the redox variability at the seabed ($S = 0.4$) and allows for oscillatory behavior. However, most of the export production and rain of particulate organic matter to the seabed is driven by large rather than small plankton in the modern ocean. **A shift to smaller plankton may, hence, decrease the oxygen demand in bottom waters** and increase rather than decrease the redox variability at the seabed.

REPLY: The pertinent bottom waters are those of the shelf seas, in which most phosphorus is buried (or recycled), not the much deeper waters of the open ocean. Changes in the efficiency of sinking are less pertinent over these much shallower depths. The ideal parameter “nutrient uptake efficiency (k_u)” determines oxygen demand in deeper water, and is under control of the cell-size of phytoplankton (Aksnes and Egge, 1991; Kriest and Oschlies, 2007):

$$k_u = \frac{\mu_1 * \left(\frac{d}{d_0}\right)^{2-\zeta} * \min(\text{NO}_3, \text{PO}_4)}{K_1 * \left(\frac{d}{d_0}\right) + \min(\text{NO}_3, \text{PO}_4)}$$

where μ_1 ($\sim 2d^{-1}$) is the maximum possible growth rate for the smallest cells of diameter d_1 ($2e-7$ m). K_1 is the half-saturation constant (e.g., $0.005 \text{ mmol N m}^{-3}$). $\min(\text{NO}_3, \text{PO}_4)$ is the ambient nutrient concentration (e.g., $0.1 \text{ mmol N m}^{-3}$). The exponent ζ (lies between ~ 2.1 to 3.0 , we take the minimum value 2.1 here) yields a decrease in cell-size, leading to an increase in nutrient uptake efficiency.

In our model, the forcing S controls the minimum nutrient uptake efficiency along the water columns, hence we substituted the maximum cell-size of the eukaryotic marine phytoplankton ($0.2\text{-}30 \mu\text{m}$) and cyanobacteria ($3\text{-}10 \mu\text{m}$) to obtain the minimum k_u . For the cell size $d = 30\mu\text{m}$ and $10 \mu\text{m}$ gives $k_u = 0.14$ and 0.38 , respectively.

Matching up well with the values of the forcing S we set here pre- and post-PTB, this also agrees with our estimate that the tipping point of S between the stable and unstable system is around 0.3 to 0.4 (Fig. 4).

The authors also state that small phytoplankton has a higher nutrient uptake efficiency than large plankton. It is, however, **more likely that P uptake efficiency increases when other limiting nutrients (N, Fe) are available such that P can be consumed by phytoplankton.** The authors' hypothesis that a shift to smaller phytoplankton is responsible for the oscillatory behavior might be valid within the framework of their simple model but is very likely not valid for the much more complex natural system. **It seems to be more likely that a high availability of N and Fe is needed to promote P uptake efficiency and oscillatory behavior.**

REPLY: Whatever is the limiting nutrient its diffusion limited uptake efficiency (per unit mass or volume) is greater for smaller cells, because they have a greater surface area to volume ratio. This uptake efficiency parameter then gets multiplied by the concentration of the limiting nutrient (whatever it is). 'P' in the model can be thought of as whatever nutrient is proximately limiting so long as it tracks variations in P on long timescales. That is what the new variant of the model with an N cycle shows. The newly added Figure S15 shows that high availability of N is not essential to sustain the **~1.5-Myr-period** limit cycle oscillation. Whereas N is limiting on short timescales ($\sim 10^3$ yr), P is limiting on longer timescales ($\sim 10^5$ yr).

The N-P-Fe dominated system or differential equations is too fast to be responsible for the oscillations during the Early Triassic, which does a great job on reproducing the **~40 kyr** period oscillation during Cretaceous (Wallmann et al., 2019).

Many thanks to the reviewer for pointing out different dynamic system with similar unstable oscillatory behaviours across different geological intervals, we have added the discussion and the comparison of the N-P-Fe and P-O-A system in the main text (See section XXX).

References

Jurikova, H., Gutjahr, M., Wallmann, K., Flogel, S., Liebetrau, V., Posenato, R., Angiolini, L., Garbelli, C., Brand, U., Wiedenbeck, M., & Eisenhauer, A. (2020, Nov). Permian–Triassic mass extinction pulses driven by major marine carbon cycle perturbations [Article]. *Nature Geoscience*, 13(11), 745–+. <https://doi.org/10.1038/s41561-020-00646-4>

Wallmann, K., Flogel, S., Scholz, F., Dale, A. W., Kemena, T. P., Steinig, S., & Kuhnt, W. (2019, Jun). Periodic changes in the Cretaceous ocean and climate caused by marine redox see-saw [Article]. *Nature Geoscience*, 12(6), 456– 462. <https://doi.org/10.1038/s41561-019-0359-x>

Wallmann, K., Jose, Y. S., Hopwood, M. J., Somes, C. J., Dale, A. W., Scholz, F., Achterberg, E. P., & Oschlies, A. (2022, May). Biogeochemical feedbacks may amplify ongoing and future ocean deoxygenation: a case study from the Peruvian oxygen minimum zone. *Biogeochemistry*, 159(1), 45–67. <https://doi.org/10.1007/s10533-022-00908-w>

REPLY: Thanks for that, all refs are added to the list of Ref.

#####

Reviewer #2 (Remarks to the Author):

The authors employed an ocean phosphorus (P), global atmosphere-ocean oxygen (O), and carbon (A) -coupled biogeochemical model to explore the periodic oscillations in isotopic signatures during the 10-million-year interval of the Early Triassic following the Permian-Triassic mass extinction. Through modeling and sensitivity analysis, these oscillations are attributed to the destabilization of coupled phosphorus, carbon, and oxygen biogeochemical cycles, driven by the homogenization of ocean floor redox conditions caused by an increase in nutrient uptake efficiency within the ocean. Overall, this is a concise manuscript. However, I am concerned that **some critical evidence supporting the authors' conclusions is missing, and additional explanations are required** to fully substantiate their claims and ensure for publication.

REPLY: We sincerely thank you for your constructive suggestions and critical comments, which significantly improve the quality of our manuscript.

The solid evidence of the bloom of the cyanobacteria during Early Triassic is added, and we have now made a new short-timescale forcing scenario that can reproduce the first U isotope excursion.

Major comments

1. The correlation with geological records remains unclear (Figure 4). While I acknowledge that the biogeochemical model indicates an unstable system resulting in periodic oscillations in carbonate carbon and uranium isotopes during the Early Triassic, the **model outputs appear to deviate from geological records**, particularly regarding uranium isotopes (Figure 4E, showing a time shift between the records and model outputs) and temperature (Figure 4D, with mismatched magnitudes at both the maximum around 252 Ma and the minimum around ~247 Ma). This discrepancy hinders confidence in determining whether adjustments to

the uptake efficiency parameters are valid.

REPLY: A new model run with short-timescale forcing scenario that can reproduce the ~252 Ma uranium isotope excursion has been added.

The maximum temperature record at ~252 Ma involves systematic difference between different locations: in South China the $\delta^{18}\text{O}_{\text{apatite}}$ changed from 22.5‰ to 18‰ across the PTB (Paleo-latitude: 20° N, Sun et al., 2012), in Iran the $\delta^{18}\text{O}_{\text{apatite}}$ changed from 19.5‰ to 16‰ (Paleo-latitude: 0°, Chen et al., 2020). The later lacks the continuous Early Triassic signals, thus, we are not modeling for the significant thermal peak signal from Iran. The minimum temperature record at around 247 Ma was interpreted to be due to the salinity offset in conodont Oxygen isotopes, which was excluded by the original paper (Sun et al., 2012). We've cleaned our oxygen isotope dataset that minimizes the discrepancy between model results and geochemical records.

The major mismatching between the model and data lies on the very first uranium isotopes anomalies at ~252 Ma. We've added short-timescale forcings (enhanced oxidative weathering, and enhanced phosphorus input to the ocean) to the original model run in main text to capture both early-Triassic oscillations and ~252 Ma $\delta^{238}\text{U}_{\text{carb}}$ excursions (see Figure S14).

Figure S14. Extended model run with short-timescale enhanced oxidative weathering ($F_{\text{oxidw}}=[-254,-252,-251.999,-251.998,-251.997,-237]$, $[1.0,1.0,100,100,1.0,1.0]$), and enhanced phosphorus flux to ocean ($F_{\text{p}}=[-254,-252,-251.99,-251.5,-251.49,-237]$, $[0,0,2e10,2e10,0,0]$) in unstable systems.

We considered two extra forcings that contribute to ~252 Ma ocean anoxic event, hence the isotope excursions: (1) massive terrestrial biomass oxidation happened during EPME (Dal Corso et al., 2020; Dal Corso et al., 2022), which decreased the atmosphere oxygen level; (2) dramatic enhancement of the delivery of nutrient (e.g., P) to the oceans through rivers due to the eruption of the Siberia Traps and widespread wildfire (Shen et al., 2011; Benton and Newell, 2014; Grasby et al., 2017), which enhanced new production and led to ocean anoxia (see details in Section 5 in SI, and Figure S14).

However, none of these short-timescale forcings can lead to Early-Triassic-long oscillations without the destabilization of the system, instead, they were suggested to have a huge impact on the very first Early Triassic geochemical anomalies. As we emphasized in the Introduction and Model limitations chapters, these short-timescale forcings complement our system-level hypothesis to establish a better fit between model and data.

Additionally, I recommend avoiding the use of light-grey color for the geological records in the figure. Clear visual distinctions between the model outputs and geological records are essential for effective comparison.

REPLY: Thanks for pointing out this, we've changed the color to dark-gray and changed the color and linewidth of the model results for a better comparison (see new Figure 4).

2. The instability of the system appears to arise from homogeneous redox conditions, with the limited range of uptake efficiency among small-celled phytoplankton (e.g., cyanobacteria) across the ocean identified as a key factor. **However, I question whether convincing**

evidence exist to support this fundamental assumption.

Cyanobacteria are typically abundant in oligotrophic marine environments, where they thrive due to smaller cell sizes and highly efficient nutrient uptake systems. This enables them to outcompete other phytoplankton. However, when nutrient levels increase and become abundant, cyanobacteria may lose this competitive advantage, and eukaryotic algae generally outcompete cyanobacteria due to their larger size, higher nutrient demands, and faster growth rates.

REPLY: This is a fair point, but we also note that in contemporary nutrient-rich eutrophic lakes, cyanobacteria dominate. There are several independent solid lines of evidence to support the crisis of the eukaryotic marine phytoplankton and the bloom of cyanobacteria during the Early Triassic. (1) Unambiguous decline in eukaryotic marine phytoplankton is evidenced by disappearance of macro-algae e.g., Dasyclad algae, which is also accompanied by the disappearance of other eukaryotes e.g., calcareous sponge and scleractinian corals during the entire Early Triassic (Knoll et al., 2007; Chen and Benton, 2012). (2) A general increase in abundance of green sulfur and N-fixing cyanobacteria is evidenced by high 2-methylhopanoid indices in basal Triassic shales (Grice et al., 2005; Xie et al., 2005, 2007; Hays et al., 2007; Cao et al., 2009; Luo et al., 2011). (3) The bloom of the cyanobacteria and other microbes is also evidenced by the widespread distributions of microbialites (stromatolites, thrombolites and other forms), and broad “anachronistic facies” or “microbially induced sedimentary structures (MISS)” (including oncoids, giant ooids, microbial mats, sand veins, wrinkle structures, vermicular limestone, flat-pebble conglomerates, cement fans etc.) worldwide (Baud et al., 1997, 2005, 2007; Pruss et al., 2006; Chen and Benton, 2012; Kershaw et al., 2012; Mata and Bottjer, 2012; Wood, 2014; Chen et al., 2014, 2019, 2022; Luo et al., 2014, 2016; Wu et al., 2022).

The oceanic nutrient level is not the only factor controlling the rise and fall of the eukaryotic marine phytoplankton or cyanobacteria. The dramatic climate change during the Early Triassic (e.g., hot-house and ocean anoxia) can directly kill the eukaryotic marine phytoplankton, while cyanobacterial autotrophs may have flourished in the aftermath of the crisis due to their greater resistance to harsh environmental conditions and a competitive advantage over eukaryotic phytoplankton in anoxic waters rich in

ammonium (Knoll et al., 2007).

This consensus appears to contradict the authors' argument, which suggests that enhanced nutrient input, driven by increased weathering, promotes the proliferation of small-celled phytoplankton (e.g. cyanobacteria).

REPLY: Our hypothesis here is not about increased nutrient input due to weathering, but the cyanobacterial flourishing as the 'disaster taxa', which increased the nutrient uptake efficiency and destabilized the P-O-A dynamic system.

Additionally, regarding the parameter S in the model simulation, are there any direct geological records or indirect evidence (e.g., statistically supported increases in burial) to support $S=0.4\pm0.1$ instead of $S=0.1\pm0.1$ during the interval between 252 Ma and 248 Ma? This distinction is crucial, as parameter adjustments should be grounded in robust and valid evidence.

REPLY: Previous studies introduced the relationship between the cell-size and nutrient uptake efficiency (k_u) as below (Aksnes and Egge, 1991; Kriest and Oschlies, 2007):

$$k_u = \frac{\mu_1 * \left(\frac{d}{d_0}\right)^{2-\zeta} * \min(\text{NO}_3, \text{PO}_4)}{K_1 * \left(\frac{d}{d_0}\right) + \min(\text{NO}_3, \text{PO}_4)}$$

where μ_1 ($\sim 2d^{-1}$) is the maximum possible growth rate for the smallest cells of diameter d_1 ($2e-7$ m). K_1 is the half-saturation constant (e.g., $0.005 \text{ mmol N m}^{-3}$). $\min(\text{NO}_3, \text{PO}_4)$ is the ambient nutrient concentration (e.g., $0.1 \text{ mmol N m}^{-3}$). The exponent ζ (lies between ~ 2.1 to 3.0 , we take the minimum value 2.1 here) yields a decrease in cell-size, leading to an increase in nutrient uptake efficiency.

In our model, the forcing S controls the minimum nutrient uptake efficiency along the water columns, hence we substituted the maximum cell-size of the eukaryotic marine phytoplankton ($0.2\text{-}30 \mu\text{m}$) and cyanobacteria ($3\text{-}10 \mu\text{m}$) to obtain the minimum k_u . For the cell size $d = 30\mu\text{m}$ and $10 \mu\text{m}$ gives $k_u = 0.14$ and 0.38 , respectively. Matching up well with the values of the forcing S we set here pre- and post-PTB, this also agrees with our estimate that the tipping point of S between the

stable and unstable system is around 0.3 to 0.4 (Fig. 4).

The timing of the S follows the geological records mentioned above e.g., the absence of the Dasyclad algae and the widespread distributions of microbialites during the PTB (~251.9 Ma) to Smithian-Spathian boundary (~248 Ma).

3. In the model simulation, the uplift external forcing remains at a high level after 252 Ma. I wonder whether this timing is well-aligned with the Siberian Traps. According to Burgess et al. (2017), the maximum age for the cessation of Siberian Traps activity is estimated at 250.2 ± 0.3 Ma. Beyond this point, it would be challenging to sustain intensified silicate weathering and the associated enhanced input of phosphorus to the oceans. Have you conducted a sensitivity analysis of the uplift forcing (U) to address this temporal discrepancy?

REPLY: The forcing Uplift that we used here does not focus on short-timescale effect, but the long-timescale (1-20 Myr) trends triggered by the emplacement of the Siberia Traps.

We added the new sensitivity tests on the forcing Uplift, with different values and decay scenarios to address your concern.

Figure S12. Sensitivity tests of the forcing Uplift. Solid red line is the original model run in the main text. Red bars represent different Uplift (0.6 to 1.0) values in the Early Triassic. Red dash and dash-dot lines represent Uplift values decay from early most Triassic (Uplift = 0.8) to 0.6 and 0.4 at 237Ma.

4. Definitions of stable and unstable. Mathematically, based on lines 248 – 250, I can understand how the inflection point occurs as the range of ku_{min} is adjusted. However, it remains difficult to quantitatively visualize the transitional conditions under which the system becomes unstable. For instance, would it be possible to include a figure illustrating the distribution of homogeneous redox conditions (e.g., anoxia extent) across 100 ocean columns in the model? Such a visualization could provide clearer insights into the transition to instability.

REPLY: Thanks, the Fig S7 and Section 3.2 in the SI answer this. The necessary condition for the appearance of the unstable limit cycle behaviour is that the marine integrated P burial becomes a non-monotonic function of the oceanic P level (Fig S7)

when the P and organic carbon burial in homogeneous ocean floor redox condition. See Section “Dynamical analysis reveals the source of unstable behaviour” (paragraph 3 and 4) for detailed description.

Figure S7. Local redox dependence combines with globally averaged oceanfloor oxygen to determine global marine phosphorus burial. (A) Per-column oceanfloor oxygen demand k_{ui} , $n_{col} = 100$, $(k_{u_{min}}, k_{u_{max}}) = (0.1, 1.0)$ and $(0.7, 1.0)$ for stable and unstable system, respectively. (B) Fraction of global marine organic carbon burial under anoxic conditions as a function of normalized marine phosphorus level at constant atmosphere-ocean oxygen and carbon. (C) Global marine phosphorus burial as a function of normalized marine phosphorus at constant atmosphere-ocean oxygen and carbon.

Minor Comments

Lines 124–127: The argument appears self-contradictory. Are you suggesting that silicate weathering has a significant impact, whereas the other two factors have less impact? Please clarify this point.

REPLY: Revised, the P weathering has three main components: Silicates fraction (80%), Carbonates fraction (14%), and Oxidative fraction (6%) (Lenton et al., 2018). Here we made the simplification that the P weathering is 100% controlled by silicate fraction.

Line 161: Could you elaborate on the meaning of "global marine burial phosphorus burial flux"? The phrasing is somewhat unclear.

REPLY: Revised, "the global integrated marine phosphorus burial flux"

Line 203: There is a typo: "He we consider..." should be corrected.

REPLY: Thanks, revised, "Here we consider..."

Lines 306 - 309: Are you referring to Figure 4 rather than Figure 3? Please verify.

REPLY: Yes, Figure 4, thanks.

REVIEWERS' COMMENTS

Reviewer #1 (Remarks to the Author):

The manuscript has been greatly improved and the response letter of the authors addresses most of the reviewer's comments. However, I still have a few comments and questions that the authors may want to address:

REPLY: We sincerely thank you for your comments that improved the manuscript.

I agree that small phytoplankton uses nutrients more efficiently than large phytoplankton. However, large phytoplankton usually sinks more rapidly than small phytoplankton. It is, thus, likely that the export efficiency decreases when the plankton ecosystem is dominated by small plankton. This effect could lead to a smaller oxygen demand at > 100 m water depth that could partially counteract the increase in oxygen demand induced by enhanced nutrient utilization in the surface ocean. The authors may want to discuss these effects of plankton size on export efficiency and oxygen demand in section "Alternative dynamical hypotheses and model limitations".

REPLY: Thanks for pointing this out. We've added this discussion in the main text. However, if we consider the declined export efficiency, then smaller cell-size would also decrease the overall flux of organic carbon burial and decrease the oxygen source.

The nitrogen cycle included in the extended model assumes that nitrogen fixation compensates for denitrification. It is not clear whether this assumption is actually valid. This is an important caveat since negative feedbacks inherent to the nitrogen cycle would strongly affect the system behavior if nitrogen fixation could not compensate for enhanced denitrification during anoxic phases. The model assumption applied by the authors also implies that nitrogen fixation rates were greatly enhanced during periods of ocean anoxia to compensate for enhanced denitrification. Is there any evidence in the geological record that supports this model outcome?

Are biomarkers for N-fixing phytoplankton and/or microbialites/MISS showing higher concentrations/abundances during anoxic compared to oxic phases? Such data would greatly support the model hypothesis. If these data are not available, the

authors may want to discuss this lack of evidence and the possible system effects of reactive nitrogen loss during anoxia in section “Alternative dynamical hypotheses and model limitations”.

REPLY: Thanks for point this out. We truly found some clues about this. Pyrite framboids (presenting dysoxic watermass) were only confined to PTB microbialites and absent in other habitats (ramp to basin and shallow, nonmicrobialite platform sections).

Indicates that microbe bloom may have stimulated dysoxic watermass and triggered the framboid growth within microbe aggregates (Chen et al., 2022 <https://doi.org/10.1029/2021GL096998>).

We've added this discussion in the main text and SI.

Reviewer #2 (Remarks to the Author):

The authors have satisfactorily addressed most of my comments, particularly regarding the geological record support and the sensitivity analysis of uplift forcing. I fully endorse publication of the manuscript in Nature Communications. The clarifications provided in response to my remarks strengthen the paper and will benefit readers. However, I recommend adding more context—at least in the supplementary materials—on how oceanic nutrient levels influence primary production and contribute to anoxic conditions. This additional information would further enhance the manuscript's clarity and robustness.

REPLY: We sincerely thank you for your comments which strengthen the paper. We highlighted the positive and negative feedback loops of the redox-sensitive P cycle in Figure 2 and its caption. And we added the detailed description of those loops in SI section 3.2.